

# The performance of a modified EWMA control chart for monitoring autocorrelated PM2.5 and carbon monoxide air pollution data

Yadpirun Supharakonsakun, Yupaporn Areepong and Saowanit Sukparungsee

Department of Applied Statistics, King Mongkut's University of Technology North Bangkok, Bangkok, Thailand

## ABSTRACT

PM2.5 (particulate matter less than or equal to 2.5 micron) is found in the air and comprises dust, dirt, soot, smoke, and liquid droplets. PM2.5 and carbon monoxide emissions can have a negative impact on humans and animals throughout the world. In this paper, we present the performance of a modified exponentially weighted moving average (modified EWMA) control chart to detect small changes when the observations are autocorrelated with exponential white noise through the average run length evaluated (ARLs) by explicit formulas. The accuracy of the solution was verified with a numerical integral equation method. The efficacy of the modified EWMA control chart to monitor PM2.5 and carbon monoxide air pollution data and compare its performance with the standard EWMA control chart. The results suggest that the modified EWMA control chart is far superior to the standard one.

Corresponding author
Yupaporn Areepong,
yupaporn.a@sci.kmutnb.ac.th

## INTRODUCTION

One of the world's greatest health and environmental problems is air pollution, which has steadily increased worldwide due to global industrialization and urbanization (*Manisalidis et al., 2020*). Air quality monitoring in many countries has shown that the levels of common pollutants have increased from 1980 until now. During 2010–2016, an estimated 55.3% of the global population were exposed to dangerous levels of air pollution (*Shaddick et al., 2020*). Therefore, a major global health risk to the world's population is air pollution, with more than 90% of people living in areas that do not meet the 2017 World Health Organization's (WHO) recommended threshold of air quality levels (*Health Effects Institute, 2019*). The WHO data also reveals that in 2016, 6.1 million deaths worldwide (around 11% of the total global deaths) are attributable to air pollution.

A widespread air pollutant consisting of a mixture of solid and liquid particles suspended in air is called particulate matter (PM), in which the physical and chemical characteristics depend on the location. The common chemical components of PM include nitrates, sulfates, ammonium, other inorganic, and biological compounds (*Huang et al., 2014*). PM

is used as an indicator that is relevant to health and is classified in terms of the diameter or width of the particles, e.g., PM10 means the diameter is 10 μm or less and PM2.5 is 2.5 μm or less (*Saarikoski et al., 2018*). PM2.5 can be human-made and naturally occurring. Air pollution from vehicle emissions and traffic congestion produces both PM2.5 and carbon monoxide (CO) pollution, with major causes being diesel engine exhaust fumes and traffic congestion (*Pui, Chen & Zuo, 2014*; *Tian et al., 2015*). Emissions from factories and power plants are also major causes of PM2.5 pollution due to burning fossil fuels, especially coal (*Hua et al., 2015*). Moreover, the burning of agricultural waste, especially in South East Asia, and forest fires are other major causes of PM2.5 pollution (*He, Lui & Zhou, 2020*; *Lee et al., 2018*).

The small size of PM2.5 particles makes them particularly deleterious to health by directly cause respiratory and cardiovascular morbidity. Since the lungs are the primary organs affected by PM2.5, exposure can lead to lung injury by perturbations of the lung microbiome and its associated metabolome mechanism (*Li et al., 2020*), as well as causing respiratory diseases and lung cancer (*Xing et al., 2016*). However, PM2.5-induced neuroinflammation and metabolic turbulence may be mitigated by the anti-inflammatory and anti-oxidative activities of fisetin (*Xu et al., 2020*). Long-term exposure to PM pollution in the air causes extrinsic skin aging, (wrinkles and changes in pigmentation). Moreover, atmospheric pollutants also lead to skin diseases such as atopic dermatitis (*Liao, Eie & Sun, 2020*).

The trend in Central and Southern Asia has been a rise in PM2.5 levels between 2010 and 2016 (*Shaddick et al., 2020*; *Cheong et al., 2019*; *Johnston et al., 2019*). Recently in Thailand, especially in Bangkok and the surrounding area, the population has been exposed to dangerously high PM2.5 pollution levels due to agricultural waste burning, foreign sources, and industry. The levels of these small particles have increased over the past few years (*Wimolwattanapun, Hopke & Pongkiatkul, 2011*; *Oanh et al., 2013*; *Pinichka et al., 2017*; *Mahidol University, 2020*). Self-care practices should be the first priority to manage this problem, while PM2.5 pollution levels should be kept under control and closely monitored periodically.

Statistical process control is a quality control approach for carrying out statistical methods to monitor and control process change. In particular, the control chart is a tool that is widely used to monitor processes to detect any changes in them, thereby preventing the occurrence of faults. In many ecological and environmental sources of data, the values are not independent over time and often comprise autocorrelated observations (autocorrelation or serial correlation is a measure of the correlation between current variable values and their past ones). Hence, the assumption in traditional control chart methodology that the observations taken from the process are independent and normally distributed does not hold. Thus, traditional control charts such as the Shewhart control chart have drastically reduced efficiency when applied to serially correlated observations. Therefore, many researchers have proposed the cumulative sum (CUSUM) (*Page, 1954*) and exponentially weighted moving average (EWMA) (*Roberts, 1959*) control charts as suitable alternatives for when the observations are autocorrelated. The EWMA control chart is recommended when the observations are not normally distributed or are autocorrelated, as has been determined using the average run length (ARL) statistic (*Srivastava & Wu,*

*1993*; *Wardell, Moskowitz & Plante, 1994*; *Jiang, Tsui & Woodall, 2000*; *Carson & Yeh, 2008*; *Han et al., 2010*). The usefulness of control chart techniques has been investigated for air pollution data in a time series. The CUSUM control chart has been employed for identifying changes in the mean air pollution level (*Barratt et al., 2007*), as well as for identifying important change-points in the time series of air pollutants measured at a busy roadside location in central London (*Carslaw, Ropkins & Bell, 2006*). The applicability of the CUSUM control chart for detecting changes in air pollutant concentrations in Delhi was investigated by *Chelani (2011)*, who found that they have been significantly increasing over time. Moreover, standard CUSUM and EWMA control charts have been used to detect a change in air pollutant time series data in Kuwait (*Al-Rashed, Al-Mutairi & Attar, 2019*); the results reveal that both procedures provided early alarms in the detection of changes in air pollutants. Afterward, a newly structure control statistic was proposed by *Patel & Divecha (2011)* and was also generalized structure of control statistic namely modified EWMA control chart. Several researchers investigated the performance of modified EWMA chart by different situations of non-normal distribution including *Aslam et al. (2017)*, *Herdiani, Fandrilla & Nurtiti (2018)* and *Noiplab & Mayureesawan (2019)*. The application of modified EWMA procedures was demonstrated using real-life samples by comparing with the existing charts. The results showed that the proposed control chart is efficient in quick detection of the out-of-control process (*Khan et al., 2018*; *Saghir, Ahmad & Aslam, 2019*; *Saghir et al., 2020*; *Aslam & Anwar, 2020*; *Supharakonsakun, Areepong & Sukparungsee, 2020*).

According to these prior studies on control chart performance, the ARL is utilized to measure the robustness of the chart. In this paper, a modified EWMA control chart, which was newly developed from the traditional EWMA procedure, is presented for monitoring and detecting small and abrupt changes in autocorrelated data by evaluating the ARL. Its performance was studied comparatively with the standard EWMA chart for detecting changes in PM2.5 and CO gas level.

## Modified EWMA chart

*Roberts (1959)* first proposed the EWMA control chart that is very effectively at detecting small process changes. This chart's design parameters are the multiples of widths of the control limit and smoothing parameter. The EWMA statistic is defined as

$$Z_t = (1-\lambda)Z_{t-1} + \lambda X_t, \tag{1}$$

where $0 < \lambda \leq 1$ is smoothing parameter and $X_t$, $t = 1, 2, 3, \dots$ isa sequence of autocorrelated observations with the starting value is the process target mean $Z_0 = \mu_0$.

The respective upper and lower control limits for the EWMA chart are

$$UCL = \mu_0 + H\sigma\sqrt{\frac{\lambda}{2-\lambda}\left[1-(1-\lambda)^{2i}\right]}, \tag{2}$$

$$LCL = \mu_0 - H\sigma\sqrt{\frac{\lambda}{2-\lambda}\left[1-(1-\lambda)^{2i}\right]}, \tag{3}$$

where $H$ is the width of the control limit and $\sigma$ is the process standard deviation. The term $\left[1-(1-\lambda)^{2i}\right]$ is close to 1 as $i$ becomes large. Hence, the respective upper and lower

control limit will approach their respective steady-state value as

$$UCL = \mu_0 + H\sigma\sqrt{\frac{\lambda}{2-\lambda}}, \tag{4}$$

and

$$LCL = \mu_0 - H\sigma\sqrt{\frac{\lambda}{2-\lambda}}. \tag{5}$$

Later, modified EWMA control chart was introduced by *Patel & Divecha (2011)*. They corrected the inertia problem caused by errors in the EWMA statistic by considering past changes as well as the latest change in the process. The modified EWMA chart is useful for detecting process changes in observations that are autocorrelated or independent normally distributed. Recently, *Khan, Aslam & Jun (2017)* proposed a new control statistic structure that developed from the modified EWMA control chart as follow:

$$Z_t = (1-\lambda)Z_{t-1} + \lambda X_t + k(X_t - X_{t-1}), \tag{6}$$

where $\lambda$ is the smoothing parameter; $X_t$, $t = 1, 2, 3, \ldots$ is a sequence of autocorrelated observations; $k$ is a constant; and the starting value is the process target mean $Z_0 = \mu_0$. It is similar to the EWMA statistic but with the last term extended.

This chart generates a false alarm when the $Z_n$ value violates the specified control limit. In general, the upper and lower control limit are respectively given by

$$UCL = \mu_0 + L\sigma\sqrt{\frac{\lambda + 2k\lambda + 2k^2}{2-\lambda}}, \tag{7}$$

and

$$LCL = \mu_0 - L\sigma\sqrt{\frac{\lambda + 2k\lambda + 2k^2}{2-\lambda}}, \tag{8}$$

where $L$ and $\sigma$ is the width of the control limit of modified EWMA procedure and process standard deviation, respectively.

## Method of evaluating ARL for moving average order *q* model

A time series is a series of data points ordered in time, and the goal of a time series analysis is usually to make a forecast of the future using time as an independent variable. A usual characteristic of a time series is autocorrelation, which is correlation between observations in the same dataset at different points in time. In other words, autocorrelated data portrays the similarity between observations as a function of the time lag between them. In a time series analysis, an MA(q) process is a common approach for modeling a univariate time series for which the error depends linearly on the current and numerous past values of the error term.

$$X_t = \mu + \varepsilon_t - \theta_1\varepsilon_{t-1} - \theta_2\varepsilon_{t-2} - \cdots - \theta_q\varepsilon_{t-q}, \tag{9}$$

where $\mu$ is the mean of the series, $\varepsilon_t$ is a white noise process assumed to be exponentially distributed, $\theta_i$ is a process coefficient, and the starting value of $\varepsilon_0 = s$ is given.

Let $L(u)$ denote the ARL for an MA(q) process with exponential white noise on a modified EWMA control chart (see Appendix A1 for the proof) that there exists only one solution of the integral equation (see Appendix A2). It is obtained by deriving a Fredholm integral equation of the second kind as follows:

$$L(u) = 1 - \frac{\lambda e^{\frac{(1-\lambda)u}{\alpha_0(\lambda+k)}}\left[e^{-\frac{b}{\alpha_0(\lambda+k)}} - 1\right]}{\lambda e^{\frac{-\mu}{\alpha_0}} \cdot e^{\frac{v+(\lambda\theta_1+\theta_1 k)s+(\lambda\theta_2+\theta_2 k)\varepsilon_{t-2}+\cdots+(\lambda\theta_q+\theta_q k)\varepsilon_{t-q}}{\alpha_0(\lambda+k)}} + e^{-\frac{\lambda b}{\alpha_0(\lambda+k)}} - 1}, \tag{10}$$

with in-control process parameter $\alpha_0$ and out-of-control process parameter $\alpha_1 > \alpha_0$.

## Numerical results

The ARLs of the explicit formulas are derived using a Fredholm integral equation of the second type, those of the numerical integral equation method are approximated using the Gauss-Legendre quadrature rule with 1,000 nodes, and the control ARL is set to $ARL_0 = 500$. The numerical approximation of the numerical integral equation and the exact result of the explicit formulas to measure the accuracy in the comparative study according to the relative error is defined as

$$\varepsilon = \frac{|L(u) - \hat{L}(u)|}{L(u)} \times 100\%, \tag{11}$$

where $L(u)$ is derived from the ARL using the explicit formulas and $\hat{L}(u)$ is an approximation of the ARL with the numerical integral equation. The numerical results are reported in Tables 1, 2 and 3.

Computation of the ARL by using the explicit formulas and the numerical integral equation method on the modified EWMA control chart were carried out with a varied smoothing parameters ($\lambda = 0.05, 0.10, 0.15$ and $0.2$); constant $k = 1$; in-control process parameter $\alpha_0 = 1$; out-of-control process parameter $\alpha_1 = (1+\delta)\alpha_0$, where $\delta$ is the shift size set as 0.001, 0.003, 0.005, 0.01, 0.05, 0.10, 0.50 or 1.00; and in-control process was $ARL_0 = 500$ (Tables 1 and 2). The parameters were set as $\mu = 2$; coefficients $\theta_1 = -0.3$, $\theta_2 = 0.5$ with $\lambda = 0.05, 0.1$ and $\theta_1 = 0.1$, $\theta_2 = 0.3$ with $\lambda = 0.15, 0.2$ for an MA(2) process; and coefficients $\theta_1 = 0.3$, $\theta_2 = 0.5$, $\theta_3 = 0.7$ with $\lambda = 0.05, 0.1$ and $\theta_1 = -0.5$, $\theta_2 = -0.3$, $\theta_3 = -0.5$ with $\lambda = 0.15, 0.2$ for an MA(3) process. The results indicate that when the smoothing parameter was increased, the value of $ARL_1$ was reduced.

The results in Tables 1 and 2 show that the analytically explicit expression of the ARL was in excellent agreement with the approximated ARL obtained from the numerical integral equation (NIE) method. The computational time for the numerical integral equation method was around 21 and 23 s for the MA(2) and MA(3) processes, respectively, while the explicit formulas required a computational time of less than one second for both.

Table 3 reports the ARL values obtained by using the explicit formulas and numerical integral equation method. The parameters were set as $\mu = 2$; coefficient parameters $\theta_1 = -0.3$, $\theta_2 = 0.7$, $\theta_3 = -0.5$ for MA(3) process; $\lambda = 0.1$; and $k$ was varies as $5\lambda$, $10\lambda$, $20\lambda$, or $50\lambda$. The results revealed that when the constant $k$ was large, $ARL_1$ was reduced.

The performances of the standard and modified EWMA control charts were also compared. These were obtained by the explicit expression when varying $\lambda$ (0.05 and 0.10)

**Table 1** The ARL of explicit formulas and NIE method for MA(2) when $\mu = 2$ and $k = 1$ on modified EWMA control chart.

| $\lambda$ | $\theta_i$ | $b$ | $\delta$ | Explicit | NIE | $\varepsilon$ |
|---|---|---|---|---|---|---|
| | | | 0.00 | 500.000070 | 500.000063 | $1.249 \times 10^{-6}$ |
| | | | 0.001 | 344.029967 | 344.029963 | $1.098 \times 10^{-6}$ |
| | | | 0.003 | 211.859210 | 211.859208 | $9.666 \times 10^{-7}$ |
| | | | 0.005 | 153.059939 | 153.059937 | $9.058 \times 10^{-7}$ |
| 0.05 | $\theta_1 = -0.3$ | 0.4528820782 | 0.01 | 90.369435 | 90.369435 | $8.348 \times 10^{-7}$ |
| | $\theta_2 = 0.5$ | | 0.05 | 21.191203 | 21.191203 | $6.871 \times 10^{-7}$ |
| | | | 0.10 | 10.915019 | 10.915019 | $5.891 \times 10^{-7}$ |
| | | | 0.50 | 2.615077 | 2.615077 | $2.141 \times 10^{-7}$ |
| | | | 1.00 | 1.693016 | 1.693016 | $8.269 \times 10^{-8}$ |
| | | | 0.00 | 500.000081 | 500.000068 | $2.634 \times 10^{-6}$ |
| | | | 0.001 | 334.507743 | 334.507737 | $1.996 \times 10^{-6}$ |
| | | | 0.003 | 201.308260 | 201.308257 | $1.479 \times 10^{-6}$ |
| | | | 0.005 | 144.002910 | 144.002908 | $1.254 \times 10^{-6}$ |
| 0.1 | $\theta_1 = -0.3$ | 0.45905302 | 0.01 | 84.171367 | 84.171366 | $1.014 \times 10^{-6}$ |
| | $\theta_2 = 0.5$ | | 0.05 | 19.612599 | 19.612599 | $6.883 \times 10^{-7}$ |
| | | | 0.10 | 10.146990 | 10.146990 | $5.686 \times 10^{-7}$ |
| | | | 0.50 | 2.508587 | 2.508587 | $1.953 \times 10^{-7}$ |
| | | | 1.00 | 1.653220 | 1.653220 | $7.259 \times 10^{-8}$ |
| | | | 0.00 | 500.000144 | 500.000105 | $7.802 \times 10^{-6}$ |
| | | | 0.001 | 334.491414 | 334.491395 | $5.543 \times 10^{-6}$ |
| | | | 0.003 | 201.328130 | 201.328122 | $3.721 \times 10^{-6}$ |
| | | | 0.005 | 144.051692 | 144.051688 | $2.934 \times 10^{-6}$ |
| 0.15 | $\theta_1 = 0.1$ | 0.572945976 | 0.01 | 84.258340 | 84.258339 | $2.104 \times 10^{-6}$ |
| | $\theta_2 = 0.3$ | | 0.05 | 19.742557 | 19.742557 | $1.116 \times 10^{-6}$ |
| | | | 0.10 | 10.274933 | 10.274932 | $8.711 \times 10^{-7}$ |
| | | | 0.50 | 2.592448 | 2.592448 | $2.893 \times 10^{-7}$ |
| | | | 1.00 | 1.709825 | 1.709825 | $1.053 \times 10^{-7}$ |
| | | | 0.00 | 500.000089 | 500.000024 | $1.309 \times 10^{-5}$ |
| | | | 0.001 | 326.638522 | 326.638494 | $8.859 \times 10^{-6}$ |
| | | | 0.003 | 192.985129 | 192.985118 | $5.595 \times 10^{-6}$ |
| 0.2 | $\theta_1 = 0.1$ | 0.583106542 | 0.005 | 137.017229 | 137.017224 | $4.225 \times 10^{-6}$ |
| | $\theta_2 = 0.3$ | | 0.01 | 79.531204 | 79.531202 | $2.809 \times 10^{-6}$ |
| | | | 0.05 | 18.554510 | 18.554510 | $1.219 \times 10^{-6}$ |
| | | | 0.10 | 9.693785 | 9.693784 | $9.119 \times 10^{-7}$ |
| | | | 0.50 | 2.508174 | 2.508174 | $2.751 \times 10^{-7}$ |
| | | | 1.00 | 1.677336 | 1.677336 | $1.014 \times 10^{-7}$ |

Notes.

Where $\lambda$ is a smoothing parameter, $\theta_i$ is a process coefficient, $b$ is UCL, $\delta$ is the shift size and $\varepsilon$ is the relative error.

for both control charts, as reported in Tables 4 and 5. The observations were from the MA(2) and MA(3) processes with $\theta_1 = -0.1$, $\theta_2 = -0.3$, and $\theta_1 = 0.7$, $\theta_2 = 0.7$, $\theta_3 = -0.1$, respectively, for $\mu = 2$, $k = 1$, and $ARL_0 = 500$. The last row is the relative mean index

**Table 2** The ARL of explicit formulas and NIE method for MA(3) when $\mu = 2$ and $k = 1$ on modified EWMA control chart.

| $\lambda$ | $\theta_i$ | $b$ | $\delta$ | Explicit | NIE | $\varepsilon$ |
|---|---|---|---|---|---|---|
| | | | 0.00 | 500.000035 | 499.999925 | $2.200 \times 10^{-5}$ |
| | | | 0.001 | 416.626140 | 416.226056 | $2.013 \times 10^{-5}$ |
| | | | 0.003 | 312.391104 | 312.391049 | $1.777 \times 10^{-5}$ |
| | $\theta_1 = 0.3$ | | 0.005 | 249.841554 | 249.841513 | $1.634 \times 10^{-5}$ |
| 0.05 | $\theta_2 = 0.5$ | 1.7145985314 | 0.01 | 166.435290 | 166.435266 | $1.437 \times 10^{-5}$ |
| | $\theta_3 = 0.7$ | | 0.05 | 45.160512 | 45.160507 | $1.073 \times 10^{-5}$ |
| | | | 0.10 | 23.579116 | 23.579114 | $9.196 \times 10^{-6}$ |
| | | | 0.50 | 5.145761 | 5.145761 | $4.015 \times 10^{-6}$ |
| | | | 1.00 | 2.933150 | 2.933150 | $1.841 \times 10^{-6}$ |
| | | | 0.00 | 500.000093 | 499.999812 | $5.633 \times 10^{-5}$ |
| | | | 0.001 | 412.884959 | 412.884760 | $4.831 \times 10^{-5}$ |
| | | | 0.003 | 306.166728 | 306.166611 | $3.847 \times 10^{-5}$ |
| | $\theta_1 = 0.3$ | | 0.005 | 243.266257 | 243.266177 | $3.265 \times 10^{-5}$ |
| 0.1 | $\theta_2 = 0.5$ | 1.790036614 | 0.01 | 160.685954 | 160.685914 | $2.495 \times 10^{-5}$ |
| | $\theta_3 = 0.7$ | | 0.05 | 43.171580 | 43.171574 | $1.322 \times 10^{-5}$ |
| | | | 0.10 | 22.563452 | 22.563449 | $1.031 \times 10^{-6}$ |
| | | | 0.50 | 5.001808 | 5.001808 | $4.065 \times 10^{-6}$ |
| | | | 1.00 | 2.881284 | 2.881284 | $1.833 \times 10^{-6}$ |
| | | | 0.00 | 500.000074 | 500.000073 | $2.124 \times 10^{-6}$ |
| | | | 0.001 | 270.407899 | 270.407899 | $1.293 \times 10^{-6}$ |
| | | | 0.003 | 140.988209 | 140.988209 | $8.213 \times 10^{-7}$ |
| | $\theta_1 = -0.5$ | | 0.005 | 95.372295 | 95.372295 | $6.522 \times 10^{-7}$ |
| 0.15 | $\theta_2 = -0.3$ | 0.10145919916 | 0.01 | 52.755839 | 52.755839 | $4.928 \times 10^{-7}$ |
| | $\theta_3 = -0.5$ | | 0.05 | 11.649752 | 11.649752 | $3.004 \times 10^{-7}$ |
| | | | 0.10 | 6.024879 | 6.024879 | $2.324 \times 10^{-7}$ |
| | | | 0.50 | 1.671266 | 1.671266 | $5.983 \times 10^{-8}$ |
| | | | 1.00 | 1.247145 | 1.247145 | $0.000 \times 10^{-7}$ |
| | | | 0.00 | 500.000023 | 500.000021 | $3.506 \times 10^{-6}$ |
| | | | 0.001 | 260.636834 | 260.636833 | $1.964 \times 10^{-6}$ |
| | | | 0.003 | 133.218384 | 133.218384 | $1.140 \times 10^{-6}$ |
| | $\theta_1 = -0.5$ | | 0.005 | 89.517777 | 89.517777 | $8.557 \times 10^{-7}$ |
| 0.2 | $\theta_2 = -0.3$ | 0.1023254883 | 0.01 | 49.241334 | 49.241334 | $5.910 \times 10^{-7}$ |
| | $\theta_3 = -0.5$ | | 0.05 | 10.884204 | 10.884204 | $3.032 \times 10^{-7}$ |
| | | | 0.10 | 5.670187 | 5.670187 | $2.293 \times 10^{-7}$ |
| | | | 0.50 | 1.630976 | 1.630976 | $6.131 \times 10^{-8}$ |
| | | | 1.00 | 1.234071 | 1.234071 | $0.000 \times 10^{-7}$ |

**Notes.**
Where $\lambda$ is a smoothing parameter, $\theta_i$ is a process coefficient, $b$ is UCL, $\delta$ is the shift size and $\varepsilon$ is the relative error.
**Table 3** The ARL of explicit formulas and NIE method for MA(3) when $\mu = 2$, $\lambda = 0.1$ and $k = 5\lambda$, $10\lambda$, $20\lambda$, $50\lambda$ on modified EWMA control chart.

| $k$ | $\theta_i$ | $b$ | $\delta$ | Explicit | NIE | $\varepsilon$ |
|---|---|---|---|---|---|---|
| | | | 0.00 | 500.000066 | 500.000043 | $4.561 \times 10^{-6}$ |
| | | | 0.001 | 413.570468 | 413.570451 | $4.083 \times 10^{-6}$ |
| | | | 0.003 | 307.119823 | 307.119812 | $3.491 \times 10^{-6}$ |
| | $\theta_1 = -0.3$ | | 0.005 | 244.081098 | 244.081090 | $3.137 \times 10^{-6}$ |
| $5\lambda$ | $\theta_2 = 0.7$ | 0.3993899124 | 0.01 | 160.988028 | 160.988024 | $2.660 \times 10^{-6}$ |
| | $\theta_3 = -0.5$ | | 0.05 | 42.170082 | 42.170082 | $1.846 \times 10^{-6}$ |
| | | | 0.10 | 21.358100 | 21.358099 | $1.552 \times 10^{-6}$ |
| | | | 0.50 | 4.092363 | 4.092363 | $6.280 \times 10^{-7}$ |
| | | | 1.00 | 2.254223 | 2.254223 | $2.573 \times 10^{-7}$ |
| | | | 0.00 | 500.000057 | 500.000050 | $1.383 \times 10^{-6}$ |
| | | | 0.001 | 322.652564 | 322.652560 | $1.028 \times 10^{-6}$ |
| | | | 0.003 | 188.777532 | 188.777531 | $7.591 \times 10^{-7}$ |
| | $\theta_1 = -0.3$ | | 0.005 | 133.436625 | 133.436624 | $6.465 \times 10^{-7}$ |
| $10\lambda$ | $\theta_2 = 0.7$ | 0.3381621032 | 0.01 | 77.030788 | 77.030787 | $5.282 \times 10^{-7}$ |
| | $\theta_3 = -0.5$ | | 0.05 | 17.693001 | 17.693001 | $3.662 \times 10^{-7}$ |
| | | | 0.10 | 9.130173 | 9.130173 | $3.023 \times 10^{-7}$ |
| | | | 0.50 | 2.283033 | 2.283033 | $1.007 \times 10^{-7}$ |
| | | | 1.00 | 1.537883 | 1.537883 | $3.251 \times 10^{-8}$ |
| | | | 0.00 | 500.000064 | 500.000060 | $7.459 \times 10^{-7}$ |
| | | | 0.001 | 259.369506 | 259.369504 | $4.634 \times 10^{-7}$ |
| | | | 0.003 | 132.360364 | 132.360364 | $3.136 \times 10^{-7}$ |
| | $\theta_1 = -0.3$ | | 0.005 | 88.971221 | 88.971221 | $2.615 \times 10^{-7}$ |
| $20\lambda$ | $\theta_2 = 0.7$ | 0.416955807 | 0.01 | 49.057165 | 49.057164 | $2.118 \times 10^{-7}$ |
| | $\theta_3 = -0.5$ | | 0.05 | 11.092459 | 11.092459 | $1.451 \times 10^{-7}$ |
| | | | 0.10 | 5.913303 | 5.913303 | $1.167 \times 10^{-7}$ |
| | | | 0.50 | 1.796095 | 1.796095 | $3.341 \times 10^{-8}$ |
| | | | 1.00 | 1.339172 | 1.339172 | $7.467 \times 10^{-9}$ |
| | | | 0.00 | 500.000067 | 500.000064 | $5.106 \times 10^{-7}$ |
| | | | 0.001 | 218.076288 | 218.076288 | $2.735 \times 10^{-7}$ |
| | | | 0.003 | 102.797634 | 102.797634 | $1.760 \times 10^{-7}$ |
| | $\theta_1 = -0.3$ | | 0.005 | 67.421706 | 67.421706 | $1.455 \times 10^{-7}$ |
| $50\lambda$ | $\theta_2 = 0.7$ | 0.763809721 | 0.01 | 36.459416 | 36.459416 | $1.174 \times 10^{-7}$ |
| | $\theta_1 = -0.5$ | | 0.05 | 8.314559 | 8.314559 | $7.938 \times 10^{-8}$ |
| | | | 0.10 | 4.565969 | 4.565969 | $6.132 \times 10^{-8}$ |
| | | | 0.50 | 1.588295 | 1.588295 | $1.889 \times 10^{-8}$ |
| | | | 1.00 | 1.253079 | 1.253079 | $0.000 \times 10^{-8}$ |

**Notes.**

Where $\lambda$ is a smoothing parameter, $\theta_i$ is a process coefficient, $b$ is UCL, $\delta$ is the shift size and $\varepsilon$ is the relative error.

**Table 4** Comparison ARL for MA(2) and MA(3) when $\mu = 2$, $(\theta_1, \theta_2) = (-0.1, -0.3)$, $(\theta_1, \theta_2, \theta_3) = (0.7, 0.7, -0.1)$ and $k = 1$ on EWMA and modified EWMA control charts using by explicit formulas.

| Shift size | MA(2) | | | |
|---|---|---|---|---|
| | $\lambda = 0.05$ | | $\lambda = 0.1$ | |
| ($\delta$) | EWMA ($h = 1.540947 \times 10^{-6}$) | Modified ($b = 0.247244692$) | EWMA ($h = 9.722515 \times 10^{-2}$) | Modified ($b = 0.2494786708$) |
| 0.00 | 500.000084 | 500.000051 | 500.000053 | 500.000045 |
| 0.001 | 491.302902 | 322.103642 | 496.493518 | 311.471395 |
| 0.003 | 474.409028 | 188.165212 | 489.573629 | 177.579957 |
| 0.005 | 458.159230 | 132.886351 | 482.775802 | 124.203548 |
| 0.01 | 420.179693 | 76.597597 | 466.298614 | 70.932841 |
| 0.05 | 216.581889 | 17.452224 | 357.212201 | 16.095125 |
| 0.10 | 101.304137 | 8.935028 | 262.720549 | 8.288535 |
| 0.30 | 9.326784 | 3.244020 | 96.241953 | 3.076188 |
| 0.50 | 2.314971 | 2.182431 | 45.489101 | 2.100245 |
| 1.00 | 1.062006 | 1.475103 | 13.178794 | 1.446133 |
| 2.00 | 1.002599 | 1.189396 | 3.916046 | 1.179423 |
| *RMI* | 3.266348 | 0.057529 | 12.408089 | 0.000000 |
| **Shift size** | MA(3) | | | |
| | $\lambda = 0.15$ | | $\lambda = 0.2$ | |
| ($\delta$) | EWMA ($h = 0.26180448$) | Modified ($b = 1.495885499$) | EWMA ($h = 0.48260591$) | Modified ($b = 1.552310613$) |
| 0.00 | 500.000023 | 500.000016 | 500.000059 | 500.000012 |
| 0.001 | 497.068024 | 389.027511 | 497.491338 | 384.835986 |
| 0.003 | 491.267689 | 269.482277 | 492.501520 | 263.538348 |
| 0.005 | 485.551042 | 206.178820 | 487.548819 | 200.434640 |
| 0.01 | 471.616385 | 129.968083 | 475.331533 | 125.474506 |
| 0.05 | 376.298194 | 33.106525 | 386.389784 | 31.756669 |
| 0.10 | 288.608113 | 17.342524 | 297.158447 | 16.662591 |
| 0.30 | 117.137775 | 6.311771 | 113.435186 | 6.117180 |
| 0.50 | 58.046820 | 4.096353 | 53.536263 | 3.995715 |
| 1.00 | 17.466862 | 2.475941 | 15.918420 | 2.438029 |
| 2.00 | 5.133009 | 1.705112 | 5.033017 | 1.691792 |
| *RMI* | 6.988635 | 0.000000 | 7.082964 | 0.000000 |

**Notes.**
Where $\lambda$ is a smoothing parameter, $b$ is UCL of the modified chart, and $h$ is UCL of the EWMA chart.

$(RMI)$ defined as

$$RMI = \frac{1}{n} \sum_{i=1}^{n} \left[ \frac{ARL_{\delta_i} - ARL_{\delta_i}^{\text{smallest}}}{ARL_{\delta_i}^{\text{smallest}}} \right], \tag{12}$$

where $ARL_{\delta_i}$ denotes the ARLs of the EWMA and modified EWMA control charts obtained via the explicit formulas for each shift size and $ARL_{\delta_i}^{\text{smallest}}$ denotes the smallest of the ARLs for each shift size.

The results in Table 4 show that when $\lambda = 0.05$, the performance of the modified EWMA control chart was better than the standard one for shift sizes of 0.001, 0.003, 0.005, 0.01,

**Table 5** Comparison ARL for MA(2) observations for PM2.5 in Thailand when $\mu = 51.163$, $(\theta_1, \theta_2) = (-0.723, -0.380)$ and $\alpha_0 = 8.90$ on EWMA and modified EWMA control charts using by explicit formulas.

| Shift size | $\lambda = 0.05$ | | $\lambda = 0.1$ | |
|---|---|---|---|---|
| $(\delta)$ | EWMA ($h = 4.2219 \times 10^{-5}$) | Modified ($b = 0.02922099$) | EWMA ($h = 0.041728$) | Modified ($b = 0.03046693$) |
| 0.00 | 500.058819 | 500.062160 | 500.040155 | 500.049594 |
| 0.001 | 499.135917 | 372.303952 | 499.464575 | 371.313138 |
| 0.003 | 497.295828 | 246.500852 | 498.315774 | 245.210289 |
| 0.005 | 495.463334 | 184.318545 | 497.170110 | 183.121388 |
| 0.01 | 490.915098 | 113.156303 | 494.319608 | 112.261092 |
| 0.05 | 456.167456 | 28.053250 | 472.201199 | 27.791251 |
| 0.10 | 416.548214 | 14.711725 | 446.185321 | 14.575755 |
| 0.30 | 292.467926 | 5.402048 | 357.803981 | 5.358236 |
| 0.50 | 208.450137 | 3.507836 | 289.541426 | 3.483141 |
| 1.00 | 94.976087 | 2.098796 | 176.853408 | 2.088303 |
| 2.00 | 24.817914 | 1.428039 | 75.347531 | 1.424175 |
| *RMI* | 22.115534 | 0.000000 | 33.559477 | 0.000000 |
| Shift size | $\lambda = 0.15$ | | $\lambda = 0.2$ | |
| $(\delta)$ | EWMA ($h = 0.211034$) | Modified ($b = 0.03171369$) | EWMA ($h = 0.305241$) | Modified ($b = 0.03296119$) |
| 0.00 | 500.037390 | 500.045701 | 500.042301 | 500.068733 |
| 0.001 | 499.529929 | 370.409082 | 499.539394 | 369.591967 |
| 0.003 | 498.516876 | 244.034940 | 498.535421 | 242.964923 |
| 0.005 | 497.506311 | 182.033100 | 497.533895 | 181.042240 |
| 0.01 | 494.990736 | 111.449184 | 495.040738 | 110.710551 |
| 0.05 | 475.411216 | 27.554328 | 475.631720 | 27.339135 |
| 0.10 | 452.240112 | 14.452853 | 452.652967 | 14.341252 |
| 0.30 | 372.239132 | 5.318641 | 373.229915 | 5.282691 |
| 0.50 | 308.827934 | 3.460822 | 310.167476 | 3.440555 |
| 1.00 | 199.849984 | 2.078817 | 210.495325 | 2.070201 |
| 2.00 | 93.953526 | 1.420680 | 95.355765 | 1.417504 |
| *RMI* | 37.060241 | 0.000000 | 37.933433 | 0.000000 |

**Notes.**
Where $\lambda$ is a smoothing parameter, $b$ is UCL of the modified chart, and $h$ is UCL of the EWMA chart.

0.05, 0.10 and 0.30, whereas for shift size of 0.50, 1.00, and 2.00, the small *RMI* of the modified EWMA chart was 0.057529 while that the *RMI* of the EWMA control chart was 3.266348. When $\lambda = 0.1$, 0.15 and 0.2, the modified EWMA control chart was more powerful than the standard one for all cases of shift size with the zero *RMI*. The results indicate that overall, the modified EWMA control chart was better than the standard one at detecting process changes.

## Application of the modified EWMA chart
PM2.5 and CO gas air pollutants are being constantly emitted, which is likely to increase over time in the winter and summer seasons. When the levels of these air pollutants are high (>50 µg/m³ for PM2.5 (https://en.wikipedia.org/wiki/Air_quality_guideline) and

>10,000 ppm for CO (https://www.airqualitynow.eu/download/CITEAIR-Comparing_Urban_Air_Quality_across_Borders.pdf)), the quality of the ambient air is unhealthy to humans. Increasing PM2.5 concentration can lead to coughing, breathing difficulties, and eye irritation and can be deadly to humans.

Table 5 contains a comparison of the ARLs of the modified and standard EWMA control charts obtained via the explicit formulas. PM2.5 and CO measurements were taken every day in January and May, respectively, 2020 by the Pollution Control Department, Thailand. There were small and abrupt changes in the PM2.5 and CO level data in the Din Daeng district of Bangkok (where there is a high volume of traffic) from measurements near a busy road (*Chuersuwan et al., 2008*). The PM2.5 and CO air pollution level data were tested for autocorrelation in the observations. The Box-Jenkins technique was applied to the two air pollution datasets to determine whether they fit forecast time series data models. The models with the lowest Akaike Information Criterion (AIC) and Bayesian Information Criterion (BIC) values were considered as optimal. Moreover, $t$-test statistics proved that the two datasets were autocorrelated. The parameter values for the MA(1) and MA(2) processes were fitted and provided 51.163 for the mean and -0.723 and -0.380 for the coefficients, respectively. The PM2.5 level was found to be significant for the MA(2) process.

The efficiency of the modified EWMA procedure was also emphasized by its performance with the CO level data for the Din Daeng district, Bangkok, Thailand. Table 6 displays the ARLs of the modified and traditional EWMA control charts. The explicit formulas were used for measuring the ARLs of the CO gas level. The data were collected every day in May 2020. The analysis for an MA autocorrelated process resulted in a mean of 1.198 and −0.662, −0.479, and −0.495 for the coefficients of the MA(1), MA(2), and MA(3) processes. The results of the PM2.5 and CO air pollutant data indicate that the modified EWMA control chart was more effective than the standard one for detecting small shifts, and so confirms that it is excellent for monitoring unusual observations with undesirable values in a timely manner for all cases of exponential smoothing parameter.

The efficacy of the control charts was visualized by plotting graphs to showcase the effective results obtained from the proposed procedure in a comparative study. Figure 1 shows that the modified EWMA control chart detected upper PM2.5 shifts at the 7th to 11th and 17th to 21st observations. On the other hand, the standard EWMA control chart only detected shifts at the 10th to 26th observations, as illustrated in Fig. 2. Figure 3 exhibits that the modified EWMA chart detected upper CO level shifts at the 12th, 25th to 26th, and 30th to 31st observations. On the contrary, the original EWMA chart only detected upper CO level shifts at the 30th to 31st observations, as shown in Fig. 4.

The modified EWMA control chart detected the upper change in PM2.5 level at the 7th observation (i.e., the 7th January), which marked the beginning of extreme changes in PM2.5 emissions at the upper level. Meanwhile, the standard EWMA control chart detected the change at the 10th observation (i.e., the 10th January). Although the CO gas level emissions were low and harmless to the human body, the performance of the modified EWMA control chart for detecting the change in CO gas emissions was exemplary.

**Table 6  Comparison ARL for MA(3) observations for CO gas in Thailand when $\mu = 1.198$, $(\theta_1, \theta_2, \theta_3) = (-0.662, -0.479, -0.495)$ and $\alpha_0 = 0.1226$ on EWMA and modified EWMA control charts using by explicit formulas.**

| Shift size | $\lambda = 0.05$ | | $\lambda = 0.1$ | |
|---|---|---|---|---|
| $(\delta)$ | EWMA $(h = 2.15351 \times 10^{-10})$ | Modified $(b = 5.6968 \times 10^{-10})$ | EWMA $(h = 4.97638 \times 10^{-9})$ | Modified $(b = 5.00361 \times 10^{-10})$ |
| 0.00 | 500.018859 | 500.018440 | 500.048462 | 500.049041 |
| 0.001 | 410.684518 | 12.373073 | 418.900893 | 11.525787 |
| 0.003 | 279.691311 | 4.269468 | 296.482942 | 4.020852 |
| 0.005 | 192.837824 | 2.664904 | 212.155442 | 2.537072 |
| 0.01 | 80.144470 | 1.544761 | 96.189807 | 1.501932 |
| 0.05 | 1.406011 | 1.004560 | 1.824931 | 1.004121 |
| 0.10 | 1.007555 | 1.000146 | 1.022682 | 1.000130 |
| 0.30 | 1.000009 | 1.000000 | 1.000051 | 1.000000 |
| 0.50 | 1.000001 | 1.000000 | 1.000005 | 1.000000 |
| 1.00 | 1.000000 | 1.000000 | 1.000000 | 1.000000 |
| 2.00 | 1.000000 | 1.000000 | 1.000000 | 1.000000 |
| *RMI* | 21.935199 | 0.000000 | 25.458726 | 0.000000 |
| Shift size | $\lambda = 0.15$ | | $\lambda = 0.2$ | |
| $(\delta)$ | EWMA $(h = 8.7735 \times 10^{-6})$ | Modified $(b = 4.45335 \times 10^{-10})$ | EWMA $(h = 9.2341 \times 10^{-4})$ | Modified $(b = 4.009478 \times 10^{-10})$ |
| 0.00 | 500.061271 | 500.067195 | 500.027847 | 500.029982 |
| 0.001 | 443.510830 | 10.806701 | 459.400446 | 10.190344 |
| 0.003 | 350.844729 | 3.810433 | 389.296622 | 3.630480 |
| 0.005 | 279.559120 | 2.428984 | 331.547146 | 2.336624 |
| 0.01 | 163.232107 | 1.465789 | 226.523241 | 1.434960 |
| 0.05 | 7.394604 | 1.003757 | 23.667633 | 1.003452 |
| 0.10 | 1.543142 | 1.000116 | 4.865620 | 1.000105 |
| 0.30 | 1.007718 | 1.000000 | 1.171529 | 1.000000 |
| 0.50 | 1.001440 | 1.000000 | 1.048075 | 1.000000 |
| 1.00 | 1.000237 | 1.000000 | 1.011600 | 1.000000 |
| 2.00 | 1.000061 | 1.000000 | 1.003766 | 1.000000 |
| *RMI* | 36.248876 | 0.000000 | 47.475014 | 0.000000 |

**Notes.**

Where $\lambda$ is a smoothing parameter, $b$ is UCL of the modified chart, and $h$ is UCL of the EWMA chart.

# DISCUSSION

The findings reveal that the modified EWMA control chart performed well for the case of smoothing parameter is greater than or equal to 0.1 due to the *RMI* of the modified EWMA chart being less than the *RMI* of the EWMA chart.

When applied to real data, the modified EWMA control chart performed excellently for detecting shifts in the PM2.5 and CO pollution levels in all cases of smoothing parameter value. The smoothing parameter value of 0.1 is recommended in applications using the modified EWMA control chart. It is a good choice as it is easier to employ and performed better than the original EWMA control chart in all situations tested.

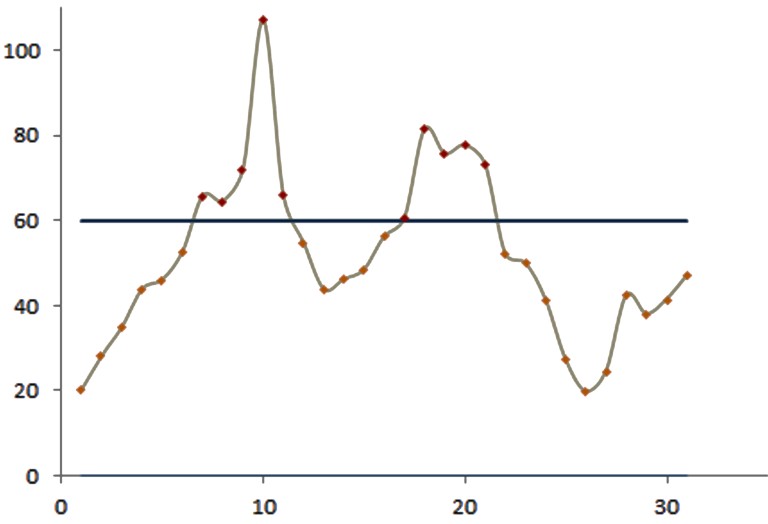

**Figure 1** The process detecting of PM2.5 level observations of modified EWMA control chart.

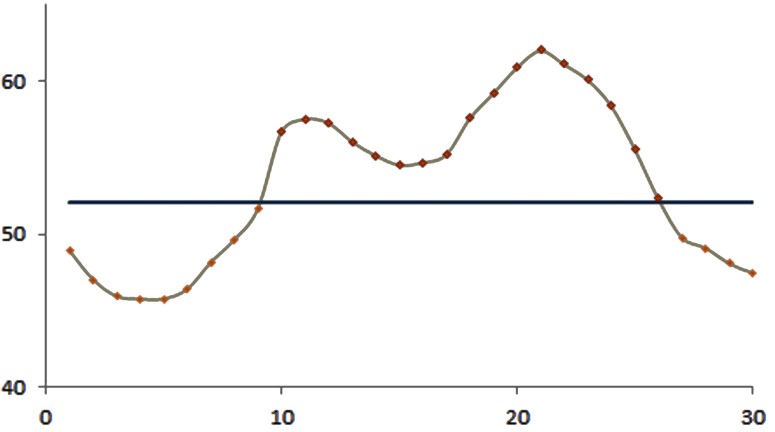

**Figure 2** The process detecting of PM2.5 level observations of the EWMA control chart.

## CONCLUSIONS

The exact ARL was provided by deriving explicit formulas that saved significantly on computational time. Therefore, it is an excellent alternative for evaluating the ARL as a measure of the effectiveness of the modified EWMA control chart. The technique showed good aptitude in monitoring and detecting small process shifts, as illustrated by changes in PM2.5 and CO gas levels examples comprising autocorrelated observations fitted to MA(2) and MA(3) models with exponential white noise. The empirical ARL shows that a smoothing parameter value of 0.1 to 0.2 supported the modified EWMA control chart far better than the EWMA control chart for all cases. Therefore, determination of the correct smoothing parameter of the chart should not be disregarded.

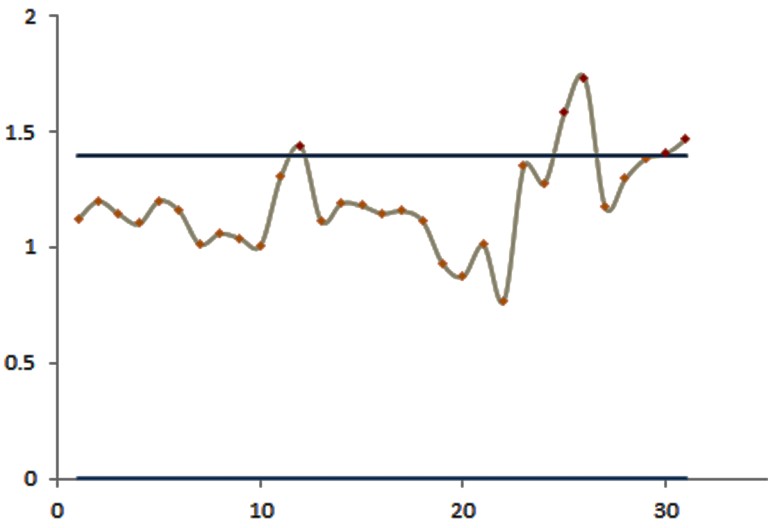

**Figure 3** The process detecting of CO gas level observations of the modified EWMA control chart.

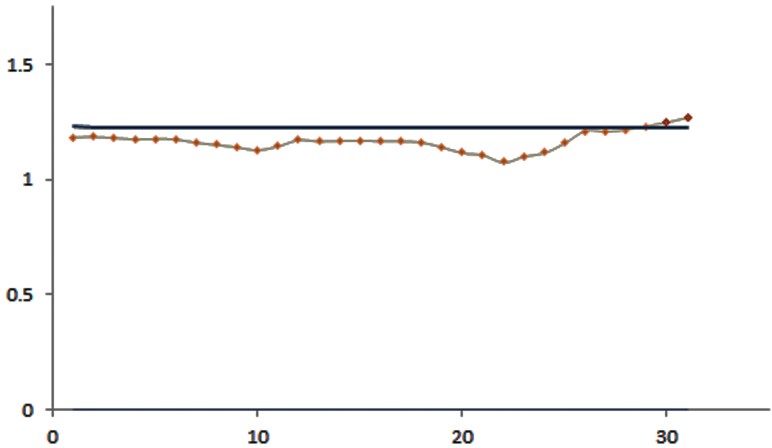

**Figure 4** The process detecting of CO gas level observations of the EWMA control chart.

## APPENDIX A1: DERIVATION ARL FOR MA($Q$) PROCESS

According to an MA($q$) process in the Eq. (9) as follows:

$$X_t = \mu + \varepsilon_t - \theta_1 \varepsilon_{t-1} - \theta_2 \varepsilon_{t-2} - \cdots - \theta_q \varepsilon_{t-q}. \tag{9}$$

Therefore, the modified EWMA statistic for an MA(q) model can be written as

$$Z_t = (1-\lambda)Z_{t-1} - X_{t-1} + (\lambda+k)\varepsilon_t + (\lambda+k)\mu - (\lambda\theta_1 + \theta_1 k)\varepsilon_{t-1}$$
$$- (\lambda\theta_2 + \theta_2 k)\varepsilon_{t-2} - \cdots - (\lambda\theta_q + \theta_q k)\varepsilon_{t-q}, \tag{13}$$

where $t = 1,2,3,\ldots$, the initial value in the process mean $Z_0 = u$, $X_0 = v$, $\varepsilon_0 = s$, and we use one side of the control limit (i.e., $LCL = 0$ and $UCL = $ b). Thus, we can obtain

$$Z_1 = (1-\lambda)u - v + (\lambda+k)\varepsilon_1 + (\lambda+k)\mu - (\lambda\theta_1 + \theta_1 k)s - \cdots - (\lambda\theta_q + \theta_q k)\varepsilon_{1-q}. \tag{14}$$

If $X_1$ causes the out-of-control state for $Z_1$, then

$(1-\lambda)u - v + (\lambda+k)\varepsilon_1 + (\lambda+k)\mu - (\lambda\theta_1+\theta_1 k)s - \cdots - (\lambda\theta_q+\theta_q k)\varepsilon_{1-q} > b$
or $(1-\lambda)u - v + (\lambda+k)\varepsilon_1 + (\lambda+k)\mu - (\lambda\theta_1+\theta_1 k)s - \cdots - (\lambda\theta_q+\theta_q k)\varepsilon_{1-q} < 0$.

If $X_1$ causes the in-control state for $Z_1$, then $0 < (1-\lambda)u - v + (\lambda+k)\varepsilon_1 + (\lambda+k)\mu - (\lambda\theta_1+\theta_1 k)s - \cdots - (\lambda\theta_q+\theta_q k)\varepsilon_{t-q} < b$.

This can be written in the form

$$\frac{-(1-\lambda)u + v - (\lambda+k)\mu + (\lambda\theta_1+\theta_1 k)s + \cdots + (\lambda\theta_q+\theta_q k)\varepsilon_{t-q}}{\lambda+k} < \varepsilon_1$$
$$< \frac{b - (1-\lambda)u + v - (\lambda+k)\mu + (\lambda\theta_1+\theta_1 k)s + \cdots + (\lambda\theta_q+\theta_q k)\varepsilon_{t-q}}{\lambda+k}.$$

The probability that $\varepsilon_1$ satisfies the bounds mentioned above for probability distribution function $\varepsilon_t$ is given as follows:

$$P\left(\frac{-(1-\lambda)u + v - (\lambda+k)\mu + (\lambda\theta_1+\theta_1 k)s + \cdots + (\lambda\theta_q+\theta_q k)\varepsilon_{t-q}}{\lambda+k} < \varepsilon_1\right.$$
$$\left. < \frac{b - (1-\lambda)u + v - (\lambda+k)\mu + (\lambda\theta_1+\theta_1 k)s + \cdots + (\lambda\theta_q+\theta_q k)\varepsilon_{t-q}}{\lambda+k}\right)$$
$$= \int_{\frac{-(1-\lambda)u + v - (\lambda+k)\mu + (\lambda\theta_1+\theta_1 k)s + \cdots + (\lambda\theta_q+\theta_q k)\varepsilon_{t-q}}{\lambda+k}}^{\frac{b - (1-\lambda)u + v - (\lambda+k)\mu + (\lambda\theta_1+\theta_1 k)s + \cdots + (\lambda\theta_q+\theta_q k)\varepsilon_{t-q}}{\lambda+k}} f(y)\,dy.$$

According to the method of *Champ & Rigdon (1991)*, let $L(u)$ denote the ARL on a modified EWMA chart for an MA(q) process. We can write the integral equation in the form

$$L(u) = 1 + \int_{\frac{-(1-\lambda)u + v - (\lambda+k)\mu + (\lambda\theta_1+\theta_1 k)s + \cdots + (\lambda\theta_q+\theta_q k)\varepsilon_{1-q}}{\lambda+k}}^{\frac{b - (1-\lambda)u + v - (\lambda+k)\mu + (\lambda\theta_1+\theta_1 k)s + \cdots + (\lambda\theta_q+\theta_q k)\varepsilon_{1-q}}{\lambda+k}} L\left[\begin{array}{l}(1-\lambda)u - v + (\lambda+k)y + (\lambda+k)\mu - \\ (\lambda\theta_1+\theta_1 k)s - \cdots - (\lambda\theta_q+\theta_q k)\varepsilon_{t-q}\end{array}\right]$$
$$f(y)\,dy. \tag{15}$$

By changing the integral variable:

$g = (1-\lambda)u - v + (\lambda+k)y + (\lambda+k)\mu - (\lambda\theta_1+\theta_1 k)s - (\lambda\theta_2+\theta_2 k)\varepsilon_{t-2} - \cdots - (\lambda\theta_q+\theta_q k)\varepsilon_{t-q}$,

we obtain

$$L(u) = 1 + \frac{1}{\lambda+k}\int_0^b L(g)f$$
$$\left(\frac{g - (1-\lambda)u + v + (\lambda\theta_1+\theta_1 k)s + (\lambda\theta_2+\theta_2 k)\varepsilon_{t-2} + \cdots + (\lambda\theta_q+\theta_q k)\varepsilon_{t-q}}{\lambda+k} - \mu\right)dg. \tag{16}$$

In this study, we define $\varepsilon_t$ is a white noise process and assumed that it is exponentially distributed with parameter $\alpha$. Therefore, the $L(u)$ is a Fredholm integral equation of the second kind as follow:

$$L(u) = 1 + \frac{1}{\lambda+k}\int_0^b L(g)\frac{1}{\alpha}e^{-\frac{g - (1-\lambda)u + v + (\lambda\theta_1+\theta_1 k)s + (\lambda\theta_2+\theta_2 k)\varepsilon_{t-2} + \cdots + (\lambda\theta_q+\theta_q k)\varepsilon_{t-q}}{\alpha(\lambda+k)} + \frac{\mu}{\alpha}}\,dg \tag{17}$$

which becomes

$$L(u) = 1 + \frac{e^{\frac{(1-\lambda)u - v - (\lambda\theta_1+\theta_1 k)s - (\lambda\theta_2+\theta_2 k)\varepsilon_{t-2} + \cdots + (\lambda\theta_q+\theta_q k)\varepsilon_{t-q}}{\alpha(\lambda+k)} + \frac{\mu}{\alpha}}}{\alpha(\lambda+k)}\int_0^b L(g)\cdot e^{-\frac{g}{\alpha(\lambda+k)}}\,dg. \tag{18}$$

Suppose that

$$C(u) = e^{\frac{(1-\lambda)u - v - (\lambda\theta_1 + \theta_1 k)s - (\lambda\theta_2 + \theta_2 k)\varepsilon_{t-2} + \cdots + (\lambda\theta_q + \theta_q k)\varepsilon_{t-q}}{\alpha(\lambda+k)} + \frac{\mu}{\alpha}}, \quad 0 \le u \le b$$

and

$$D = \int_0^b L(g) \cdot e^{-\frac{g}{\alpha(\lambda+k)}} dg, \quad where D\, is\, a\, constant.$$

Thus, we can obtain

$$L(u) = 1 + \frac{C(u)}{\alpha(\lambda+k)} D. \tag{19}$$

The ARL for an MA(q) process on a modified EWMA control chart is obtained by deriving a Fredholm integral equation of the second kind as follows:

$$L(u) = 1 - \frac{\lambda e^{\frac{(1-\lambda)u}{\alpha_0(\lambda+k)}} \left[ e^{-\frac{b}{\alpha_0(\lambda+k)}} - 1 \right]}{\lambda e^{\frac{-\mu}{\alpha_0}} \cdot e^{\frac{v + (\lambda\theta_1 + \theta_1 k)s + (\lambda\theta_2 + \theta_2 k)\varepsilon_{t-2} + \cdots + (\lambda\theta_q + \theta_q k)\varepsilon_{t-q}}{\alpha_0(\lambda+k)}} + e^{-\frac{\lambda b}{\alpha_0(\lambda+k)}} - 1}, \tag{10}$$

with in-control process parameter $\alpha_0$ and out-of-control process parameter $\alpha_1 > \alpha_0$.

## APPENDIX A2: EXISTENCE AND UNIQUENESS OF ARLS

The Banach's Fixed-point Theorem is used to show the exists and a uniqueness of the solution for ARL using the integral equation for the explicit formulas.

Let $M \ne \varnothing$ be a complete metric space and $d : M \times M \to R.d$ is a distance function on $M$ such that the following axioms hold.

1. $d(x,y) \ge 0$ for all $x,y \in M$
2. $d(x,y) = 0$ if and only if $x = y$
3. $d(x,y) = d(y,x) \leftrightarrow x = y$ for all $x,y \in M$
4. $d(x,y) \le d(x,z) + d(z,y)$ for all $x,y,z \in M$.

Since $(M,d)$ is a complete metric space, it denoted the space of all continuous function on $[0,b]$ with the norm $\|\cdot\|_\infty = \sup_{u \in I} |L(u)|$ and every Cauchy's sequence $L_{nn \ge 0}$ for a point of $M$ converges to a point $L_0 \in [0,b]$. In this case, let $T$ be an operation in the class of all continuous functions defined by,

$$T(L(u)) = 1 + \frac{1}{\lambda+k} \int_0^b L(g) \frac{1}{\alpha} e^{-\frac{g - (1-\lambda)u + v + (\lambda\theta_1 + \theta_1 k)s + (\lambda\theta_2 + \theta_2 k)\varepsilon_{t-2} + \cdots + (\lambda\theta_q + \theta_q k)\varepsilon_{t-q}}{\alpha(\lambda+k)} + \frac{\mu}{\alpha}} dg. \tag{20}$$

By Banach's Fixed-point Theorem, if an operator $T$ is a contraction, then the fixed-point equation $T(L(u)) = L(u)$ has a unique solution. The Eq. (20) exists and has a unique solution according to the following the theorem.

**Theorem 1.** (Banach's Fixed-point Theorem)

Let $(M,d)$ be a complete metric space and $T : M \to M$ be a contraction mapping with contraction constant $c \in [0,1)$ such that $\|T(L_1) - T(L_2)\| \le c\|L_1 - L_2\|$ for all $L_1, L_2 \in M$. Subsequently, there exists a unique $L(.) \in X$ such that $T(L(u)) = L(u)$ (*Sofonea, Han & Shillor, 2006*).

*Proof*. The inequality
$\|T(L_1) - T(L_2)\| \le c \|L_1 - L_2\|$ for all $L_1, L_2 \in [0, b]$ with $0 \le c < 1$.
Consider Eq. (14),

$$\|T(L_1) - T(L_2)\| = \sup_{u \in [0,b]} |L(u)|$$

$$= \sup_{u \in [0,b]} \left| (L_1(g) - L_2(g)) \frac{1}{\lambda+k} \int_0^b L(g) \frac{1}{\alpha} \right.$$

$$\left. e^{-\frac{g-(1-\lambda)u+v+(\lambda\theta_1+\theta_1 k)s+(\lambda\theta_2+\theta_2 k)\varepsilon_{t-2}+\cdots+(\lambda\theta_q+\theta_q k)\varepsilon_{t-q}}{\alpha(\lambda+k)} + \frac{\mu}{\alpha}} dg \right|$$

$$\le \sup_{u \in [0,b]} \left| \|L_1 - L_2\|_\infty \frac{-\alpha(\lambda+k)}{\alpha(\lambda+k)} e^{\frac{(1-\lambda)u-v-(\lambda\theta_1+\theta_1 k)s-(\lambda\theta_2+\theta_2 k)\varepsilon_{t-2}-\cdots-(\lambda\theta_q+\theta_q k)\varepsilon_{t-q}}{\alpha(\lambda+k)} + \frac{\mu}{\alpha}} \left[ e^{\frac{-b}{\alpha(\lambda+k)}} - 1 \right] \right|$$

$$= \|L_1 - L_2\|_\infty \left| 1 - e^{\frac{-b}{\alpha(\lambda+k)}} \right| \sup_{u \in [0,b]} \left| e^{\frac{(1-\lambda)u-v-(\lambda\theta_1+\theta_1 k)s-(\lambda\theta_2+\theta_2 k)\varepsilon_{t-2}-\cdots-(\lambda\theta_q+\theta_q k)\varepsilon_{t-q}}{\alpha(\lambda+k)} + \frac{\mu}{\alpha}} \right|$$

$$= c \|L_1 - L_2\|_\infty,$$

where $c = \left| 1 - e^{\frac{-b}{\alpha(\lambda+k)}} \right| \sup_{u \in [0,b]} \left| e^{\frac{(1-\lambda)u-v-(\lambda\theta_1+\theta_1 k)s-(\lambda\theta_2+\theta_2 k)\varepsilon_{t-2}-\cdots-(\lambda\theta_q+\theta_q k)\varepsilon_{t-q}}{\alpha(\lambda+k)} + \frac{\mu}{\alpha}} \right|$; $0 \le c < 1$ and $c$
is a positive constant.

By Theorem 1, Banach's Fixed-point Theorem guarantees the existence and uniqueness of the solution for the ARL.

### Funding
This research was funded by King Mongkut's University of Technology North Bangkok Contract no. KMUTNB-62-KNOW-018. The funders had no role in study design, data collection and analysis, decision to publish, or preparation of the manuscript.

### Grant Disclosures
The following grant information was disclosed by the authors:
King Mongkut's University of Technology North Bangkok: KMUTNB-62-KNOW-018.

### Competing Interests
The authors declare there are no competing interests.

### Author Contributions
- Yadpirun Supharakonsakun conceived and designed the experiments, performed the experiments, analyzed the data, prepared figures and/or tables, and approved the final draft.
- Yupaporn Areepong and Saowanit Sukparungsee conceived and designed the experiments, analyzed the data, authored or reviewed drafts of the paper, and approved the final draft.

### Data Availability
Raw data, including PM 2.5 and CO air pollutant data, and code are available as Supplemental Files.

## Supplemental Information

Supplemental information for this article can be found online at http://dx.doi.org/10.7717/peerj.10467#supplemental-information.

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
