# Peer review of "The performance of a modified EWMA control chart for monitoring autocorrelated PM2.5 and carbon monoxide air pollution data"

_PeerJ, doi:10.7717/peerj.10467_

## Round 0.1 · original submission · Major Revisions

· Academic Editor

Major Revisions

I agree with the reviewers that the paper would benefit from considerable restructuring - push the theoretical work to an Appendix and highlight the implications and applications.

Reviewer 1 ·

Basic reporting

The English language should be improved to ensure that an international audience can clearly understand your text. For example, in line 82, the sentence “Xt is tth observation” is not correct in English grammar.

Experimental design

The question research well defined. Research methodology is correct and there are some isuues which are not clear.

Validity of the findings

1. In the introduction part, there are no the literature reviews about the using CUSUM and EWMA charts with PM2.5 and carbon monoxide air pollution data. The literature reviews in this matter should be added in the introduction.
2. Why the author select a modified EWMA chart with the PM2.5 and carbon monoxide air pollution data? Could you give the reason?
3. In line 82, the range of parameter is not correct.
4. In the upper and lower control limits for the EWMA chart, the formula of H value or the method for finding the H value should be explained.
5. In the upper and lower control limits for the modified EWMA chart, the formula of L value or the method for finding the L value should be explained.
6. In the upper and lower control limits for the EWMA and modified EWMA charts, what is the criteria for setting the value of the smoothing parameter ?
7. In Eq.6, how do you set the value of k? Explain more in details.
8. In line 112, “UCL = b” Is it correct? What is the value of b?
9. In line 131, the details of M are not appear. (It is showed in line 137)
10. In line Eq.19, the typing of the formula is mistake.
11. In application part, the ACF and PACF for fitting the MA process should be shown and analyzed in the paper.
12. The estimation method for estimating the parameters of MA(3) process should be mentioned.

Additional comments

This paper proposed new knowledge and apply to a real data set. It is suitable for publication in your journal. There are some isuues which is not clear.

Reviewer 2 ·

Basic reporting

The manuscript proposes a modification of the EWMA control chart for monitoring air quality (PM2.5 and CO). The application is based on data collected in Thailand during 2020. The topic is relevant, interesting and it could potentially be a good application or a case study. The authors have submitted some codes and the dataset.

However, I found that the article is poorly written, the sections are not well connected and it fails to put in context this new approach. This manuscript would need a lot of work, heavy rewriting and copyediting to make it publishable.
From a practical point of view, some of these results would be very hard to reproduce. The authors should keep practitioners in mind.

I am listing here some of the main issues but not all of them.

The introduction should highlight the importance of air quality and in particular monitoring the concentrations of PM2.5, but there is not a proper literature review.
For example, in L27-39 the authors introduce what is PM2.5 and the main sources of contamination. This needs proper referencing to the relevant literature.
In L51-55 the same. The authors make some claims no properly justified with evidence/citation e.g. : “the population has been exposed to dangerously high PM2.5 pollution levels”

L64 This sentence is not grammatically correct:
“Thus, if the Shewhart control chart is applied to serially correlated
observations, the performance in detection of the process change will be poor due to the sequentiality of the numerical data values recorded for a variable over a specified period of time with data points recorded on a regular basis.”

What you are trying to say here is that Shewhart control charts assume randomness in the data.

L73 What studies are the authors referring to?

Experimental design

The presentation of the statistical methods is inadequate. For example, in Line 78 the authors attempt to describe the EWMA control chart, which is basic in statistical quality control technique. The description is ambiguous and not statistically sound e.g. “This chart design parameters are the multiple of width of control limit and smoothing parameter”. You could say that the statistic is obtained using a smoothing parameter lambda and it depends on previous realizations.

Some of the derivations and proofs could go into the appendix. For example from L113 to 161 the authors could leave what is relevant for the reader.

With so many issues in this section, is difficult to assess if there are methodological flaws too.

In L102: Here again, this is inadequate e.g. “In time series analysis, the moving average process (MA) is one of the common approach for modeling univariate time series.”
L115: What do you mean by “Change the variable, we obtain”? Please clarify

Validity of the findings

The application or case study is vaguely presented.
Why the PM2.5 and CO were measured at different time points? January the PM2.5 and May the CO. Aren't these variables correlated?
What is the justification for selecting only 31 data points? The study would be more robust if the time series was larger.
L237 Do you mean > 50?

Reviewer 3 ·

Basic reporting

.

Experimental design

.

Validity of the findings

.

Additional comments

A modified exponentially weighted moving average control chart is proposed to detect small changes when the observations are autocorrelated with exponential white noise. The performance of a modified exponentially weighted moving average (modified EWMA) control chart is illustrated through the average run length evaluated by explicit formulas. The solution accuracy is verified with a numerical integral equation method. The efficacy of the modified EWMA control chart to monitor PM2.5 and carbon monoxide air pollution data is compared with the standard EWMA control chart.

Many descriptions of the manuscript are not clear. The comments are the following.

Comments:
1. Why do the modified EWMA consider only Xt-1 not Xt-1, Xt-2,….?
The reason should be given to convince readers, although its performance looks better.
In fact, the review does not feel it is a good approach, and it is not new.
2. What is the distribution of X? The authors did not mention it. However, it is important.
3. Eq. 12 should be expressed clearly.
4. Why are the control limits symmetric? The distribution of X is not mentioned.
If it is normal distribution, then the control chart is not new.
5. In the numerical example, no distribution of the data is described or tested.
What is the AR(q) model? Does the author make the model adequate checking?
6. What are the in-control and out-of-control data for pm 2.5 and CO gas data?
7. So far, many modified EWMA control charts have been discussed in the literature.
The modified EWMA control chart should compare with the out-of-control detection speed with the existing EWMA control charts.

---

## Round 0.2 · Major Revisions

· Academic Editor

Major Revisions

Please complete the edits recommended, specifically relegating technical work (eg lines 130-159) to the Appendix. Also please note that X is a random variable and therefore has a distribution, if you are making no assumption about the distribution then make that explicit. Please clearly identify the innovation in your proposed approach. Finally, please enhance the readability of your tables by including fewer decimal places. If the point you want to make is that the Explicit and NIE columns are similar then make it another way. Please define all columns in the table caption.

---

## Round 0.3 · Major Revisions

· Academic Editor

Major Revisions

Please address the current and previous recommendations of the third reviewer.

Reviewer 1 ·

Basic reporting

The english grammar in this paper has been improved already.

Experimental design

Some mistakes have been corrected.

Validity of the findings

no comment

Additional comments

This paper is suitable for publishing in the journal.

Reviewer 3 ·

Basic reporting

.

Experimental design

.

Validity of the findings

.

Additional comments

A modified exponentially weighted moving average control chart is proposed to detect small changes when the observations are autocorrelated with exponential white noise. The performance of a modified exponentially weighted moving average (modified EWMA) control chart is illustrated through the average run length evaluated by explicit formulas. The solution accuracy is verified with a numerical integral equation method. The efficacy of the modified EWMA control chart to monitor PM2.5 and carbon monoxide air pollution data is compared with the standard EWMA control chart.
This is an application of using the EWMA chart with a time series model in air pollution data. It looks that the detection performance of the time series model and modified EWMA control chart is quite good compared to the traditional EWMA control chart. It shows a good application in air pollution data analysis.
However, the modified EWMA control chart is not new. There are many related papers about the modified EWMA control chart, some may be more effective in abnormal detection. The authors should cite them in the manuscript before the manuscript can be published. Furthermore, the distribution of error in the time series model is not normal but exponential distribution. It should be explained in a revised manuscript. Under the exponential error term, the EWMA control chart should have asymmetric control limits, not symmetric control limits.
This revised manuscript is not completely responsive to the referees’ comments. For easy reading by referees, usually, a revised manuscript should reply to referees’ comments point by point.
The reviewer suggests accepting the revised paper before the above suggestions can be considered in the manuscript.

---

## Round 0.4 · Minor Revisions

· Academic Editor

Minor Revisions

This is a much improved manuscript, thanks for working to improve it. A few small problems remain. You still have not defined all the columns in your tables. For example, what is epsilon in Table 1? And I am still not convinced that direct comparison of lists of numbers is an efficient way to demonstrate that NIE is close to Explicit.

---

## Round 0.5 · Minor Revisions

· Academic Editor

Minor Revisions

Thanks for clarifying the definitions of the column titles of the tables. Please now include that information in the captions themselves as well.

---

## Round 0.6 · accepted · Accept

· Academic Editor

Accept

I don't think that the caption will be under the table as well as over it, but the type-setters will sort it out.